# Evaluation and Validation of Colloidal Gold Immunochromatographic Qualitative Testing Products for the Detection of Emamectin Benzoate, Isocarbophos, and Fipronil in Cowpea Samples

**DOI:** 10.3390/foods14030478

**Published:** 2025-02-02

**Authors:** Anning Song, Miao Wang, Yongxin She, Maojun Jin, Zhen Cao, A. M. Abd El-Aty, Jing Wang

**Affiliations:** 1Institute of Quality Standardization & Testing Technology for Agro-Products, Chinese Academy of Agricultural Sciences, Beijing 100081, China; songanning1997@163.com (A.S.); 0891syx@163.com (Y.S.); katonking@163.com (M.J.); caozhen01@caas.cn (Z.C.); 2Key Laboratory of Agrofood Safety and Quality (Beijing), Ministry of Agriculture and Rural Areas, Beijing 100081, China; 3Department of Pharmacology, Faculty of Veterinary Medicine, Cairo University, Giza 12211, Egypt; abdelaty44@hotmail.com; 4Department of Medical Pharmacology, Medical Faculty, Ataturk University, Erzurum 25240, Turkey

**Keywords:** food safety, colloidal gold immunochromatographic qualitative testing, cowpea, pesticide residue, emamectin benzoate, isocarbophos, fipronil

## Abstract

Pesticide residues still pose a risk to human health. With the rapid development of rapid testing technology, the levels of different types of pesticide residues in agricultural products can be identified in a shorter period; thus, the safety of food can be guaranteed. However, the effectiveness of commercially available testing products has yet to be evaluated. In this study, colloidal gold immunochromatographic qualitative testing products manufactured by 34 companies were tested for their assay performance on Emamectin Benzoate, Isocarbophos, and fipronil with standardized cowpea samples. The results indicated that most of the evaluated products were identified as having ‘passed’. Most pesticide residue rapid test immunoassay products can be considered ideal means for testing certain pesticide residues. However, further evaluation of pesticide residue rapid test immunoassay products is needed, as detection technologies are still developing.

## 1. Introduction

In recent years, under the guidance of official national policies and regulations, an increasing number of enterprises/institutions and research organizations have become dedicated to developing and manufacturing rapid testing products for pesticide residues, considering these techniques’ significantly greater overall efficiency (including time, labor, and cost) than traditional HPLC/GC and HPLC/MS methods. Various rapid-testing technologies have flourished, and a multitude of rapid-detection products have emerged on the market. However, the research and development levels and product quality of manufacturers are highly varied. The establishment of more industry standards has become critical for higher-efficiency surveillance. Additionally, potential issues regarding the accuracy and reliability of rapid-testing products may exist, as verification protocols are still being developed. The quality control systems also need more improvements to achieve greater uniformity and stability in terms of product quality. Given these problems, establishing a mature evaluation standard for rapid-testing products is necessary to promote the healthy development of the rapid-testing industry through evaluation systems. To ensure food safety and regulate the rapid-testing-product market, several mature rapid-testing-product evaluation procedures have been proposed around the world, and they can be good references for the improvement of current standards and regulations. For example, the International Organization for Standardization (ISO) published “ISO 16140:2016 Microbiology of food and animal feeding stuffs—Protocol for the validation of alternative methods”. In accordance with the ISO 16140:2016 standard, supplementary amendments have been made to evaluate rapid-testing methods [1]. The Association Française de Normalization (AFNOR), the Nordic Validation System (NordVal), and the American Association of Analytical Communities (AOAC) have also published similar documents for the validation of alternative microbiological analysis methods [2,3,4]. 

The cowpea (*Vigna unguiculata* [L.] Walp.), a typical and important food crop in China, is widely cultivated in Southern China, including in Hainan Province. It has high nutritional value and is frequently included in the diets of Chinese people [5]. Owing to the relatively high temperature and humidity of cowpea cultivation areas, the occurrence of plant diseases and pests has become a problem [6]. Furthermore, it is widely acknowledged that the cowpea has a shared period of flowering and harvesting, which means that the incidence of pests and plant diseases is greater than that for some other crops [7]. For a better cowpea yield, farmers tend to apply a variety of pesticides to their crops, and emamectin benzoate is a commonly used pesticide in this regard. It has been proven to be effective against multiple species of pests, such as thrips and pod borers [8]. Although the use of emamectin benzoate is regulated, one can easily exceed the maximum residue limit (MRL), which is 0.2 mg/kg according to the GB 2763 national standard, through inappropriate handling [9]. Unlike emamectin benzoate, isocarbophos and fipronil are prohibited in the cowpea cultivation industry because of their high toxicity and potential to trigger long-term negative effects on the environment [10,11]. However, it is still possible for people to apply them to cowpea crops intentionally or unintentionally. Therefore, testing is necessary to ensure their absence in these crops.

Among all the rapid-testing products for pesticide residues, colloidal gold immunochromatographic testing products are ideal for rapid onsite screening because of their simplicity, high efficiency, low cost, minimal potential for pollution, and strong specificity. This study provides a novel approach to validating rapid pesticide residue detection methods, specifically targeting emamectin benzoate, isocarbophos, and fipronil in cowpea plants. The findings contribute to the field by establishing a standardized validation protocol and offering insights into improving the quality and reliability of rapid-testing products. These advancements are expected to have significant implications for food safety monitoring and regulatory practices.

## 2. Materials and Methods

### 2.1. Materials

The raw materials of the target pesticides were obtained from Alta Scientific Co., Ltd., Tianjin, China. Detailed information on the pesticides is shown in Table 1 and Figure 1.

The cowpea samples were obtained from Beijing Webio Reference Material Technology Co., Ltd. (Beijing, China). The instrumental analysis service (confirmatory test) for the cowpea samples was provided by Shandong Jienuo Testing Service Co., Ltd. (Weifang, China) (Report ID: A202407260085). The target pesticides (emamectin benzoate, isocarbophos, and fipronil) were not detected according to the report (not detected means that the test result was less than the limit of quantitation (LOQ)).

The standard solutions were prepared from the aforementioned raw materials of the target pesticides, providing target pesticide samples of specified concentrations.

### 2.2. Testing Products

The products being tested were colloidal gold immunochromatographic test strips. These products were obtained from a total of 34 different biotechnology companies/manufacturers in China. These companies/manufacturers were coded as T01 to T34. Among them, 29 companies’/manufacturers’ products were tested for emamectin benzoate, 31 companies’/manufacturers’ products were tested for isocarbophos, and 33 companies’/manufacturers’ products were tested for fipronil.

The information documents on the product were gathered from all the companies/manufacturers that took part in the test. The documents included instruction manuals, inspection reports, technical principles, and manufacturing process materials of the products for each kind of pesticide. The technical indicators of the rapid test products were confirmed to be consistent with those stated in the documents submitted by each company. The enterprise qualifications (including business licenses, licenses for medical device production, and ISO system certifications) of each company/manufacturer were also collected along with these information documents.

The evaluated products employ the specific antigen–antibody reaction in combination with lateral flow chromatography and colloidal gold technology for the rapid qualitative detection of the three pesticide residues in samples. During the detection process, pesticide residues in the sample bind to the colloidal-gold-labeled specific monoclonal antibodies during flow, thereby inhibiting the binding of the antibodies to the haptene–protein conjugates on the nitrocellulose membrane test line (T line). This inhibition results in a change in the coloration of the test line. By comparing the intensity of the coloration between the test line (T line) and the control line (C line), the presence of the pesticide residue in the sample can be qualitatively determined.

### 2.3. Sample Preparation

Prior to subdivision, the samples were weighed according to the operating requirements specified in each of the product instruction manuals after being verified via high-performance liquid chromatography–tandem mass spectrometry, ensuring the absence of relevant pesticides in the samples. Notably, the masses of the samples required by each product may differ slightly.

The blank samples were divided into 84 equal portions, with 21 portions serving as blank samples and the remaining 63 portions used to prepare spiked samples. The 63 blank samples were subsequently divided into 3 groups, each containing 21 portions. The samples in each group were spiked with standard solutions of 0.25 times the MRL (0.25 MRL), the MRL, and 2 times the MRL (2 MRL) according to the MRLs specified in the GB 2763 standard. The MRLs of the three target pesticides in cowpea are shown in Table 1. The samples were spiked 4 h before the product-testing procedure according to the order of testing.

After the samples were spiked, they were randomly coded via random code generation software. This procedure imparts a unique number to each sample with the corresponding concentrations of added pesticide solution. The sample-grouping and -encoding rules are shown in Table 2.

The setup of 21 samples per group was basically designed in reference to the European Union’s guidelines for the validation of testing methods for veterinary drug residues, which includes setting up 20 blank samples, as well as the original Ministry of Agriculture’s reference evaluation criteria for the filing of veterinary drug residue test kits, which stipulates that 20 blank samples and 20 MRL-spiked samples should be used to calculate the detection limit and critical value [12,13].

In this study, a sample size of 21 was deemed to generally meet the requirements of the evaluation test. When there is 1 nonconforming sample (false positive or false negative), the false-positive or false-negative rate is less than 5%. When there are 2 nonconforming samples (false positives or false negatives), the false-positive or false-negative rate is less than 10%, which facilitates the statistical analysis and evaluation of the results.

### 2.4. Experimental Process

As soon as the samples were ready, test strip cards were obtained from each product by laboratory technicians. This process was carried out strictly according to the guidance provided in the product instruction manual, observing the color reactions and making judgments on the results. The sample numbers for those that tested positive and negative were recorded, and photos of the test strip cards were taken in a timely manner for record-keeping (when the photos were taken, the test strip cards were arranged in an orderly manner). The test results and other information were entered into the formal results collection form. The photos were also printed and attached to the resulting collection form.

### 2.5. Assay Performance and Evaluation Criteria

After the experimental process had ended, the reviewing experts conducted calculations according to Table 3 and filled the results into the result collection form. All original data, calculations, and product instruction manuals were signed and confirmed upon completion of the test before being handed over to the assessment team for future review.

A product’s evaluation result was ‘pass’ only when all the requirements in Table 4 were met. Otherwise, the result was ‘fail’.

## 3. Results

The results for the three target pesticides can be found in Appendix A (Table A1, Table A2 and Table A3 and Figure A1, Figure A2 and Figure A3).

### 3.1. Emamectin Benzoate

Among the products tested for emamectin benzoate, 26 out of 29 (89.7%) passed the evaluation, leaving 3 products categorized as having failed. For the 26 products that passed, the analysis revealed a 4.76% false-positive rate (1 out of 21 samples) in the blank samples for those from T32. Similarly, a 4.76% false-positive rate (1 out of 21 samples) was observed for the 0.25 MRL samples from products by T29, T30, and T33. Additionally, a 4.76% false-negative rate (1 out of 21 samples) for the MRL samples was identified for products from T32 and T33.

Among the three products that failed, the product from T20 had a 14.28% false-positive rate (3 out of 21 samples) for the blank samples, a 4.76% false-positive rate (1 out of 21 samples) for the 0.25 MRL samples, and a 19% false-negative rate (4 out of 21 samples) for the MRL samples. The T25 product presented a 4.76% false-positive/-negative rate (1 out of 21 samples) across all four sample groups. Moreover, the product from T28 demonstrated a 4.76% false-positive rate (1 out of 21 samples) for both the blank and 0.25 MRL samples, alongside a 9.52% false-negative rate (2 out of 21 samples) for the MRL samples.

### 3.2. Isocarbophos

For isocarbophos, 30 of the 33 products (90.9%) were identified as passing products, whereas three failed. Among the passing products, a 4.76% false-positive rate (1 out of 21 samples) was recorded for blank samples from T29. A similar false-positive rate for 0.25 MRL samples was identified for products from T6, T17, and T22, whereas T26 products presented a higher (9.52%) false-positive rate (2 out of 21 samples) for the 0.25 MRL samples. Additionally, a 4.76% false-negative rate (1 out of 21 samples) for the MRL samples was observed for products from T17, T26, and T29.

Among the products that failed, the product from T30 had a 9.52% false-positive rate (2 out of 21 samples) for the blank samples and an equal false-negative rate (2 out of 21 samples) for the MRL samples. The T31 product presented a significant 33.3% false-positive/negative rate (7 out of 21 samples) for the blank and MRL samples, along with a 9.52% false-positive/negative rate (2 out of 21 samples) for the 0.25 MRL and 2 MRL samples. The T33 product had a 9.52% false-positive/negative rate (2 out of 21 samples) for the blank and 2 MRL samples, a striking 28.6% false-positive rate (6 out of 21 samples) for the 0.25 MRL samples, and a 23.8% false-negative rate (5 out of 21 samples) for the MRL samples.

### 3.3. Fipronil

For fipronil, 27 of the 31 products (87.1%) passed the tests, whereas four failed. Among the passing products, the T33 product presented a 4.76% false-positive/negative rate (1 out of 21 samples) for the blank and MRL samples.

Among the products that failed, the product from T12 had a 47.6% false-positive rate (10 out of 21 samples) for the blank samples, a 38.1% false-positive rate (8 out of 21 samples) for the 0.25 MRL samples, a 52.4% false-negative rate (11 out of 21 samples) for the MRL samples, and a 23.8% false-negative rate (5 out of 21 samples) for the 2 MRL samples. The T22 product presented a 4.76% false-positive rate (1 out of 21 samples) for the blank samples, a 23.8% false-positive rate (5 out of 21 samples) for the 0.25 MRL samples, and a 28.6% false-negative rate (6 out of 21 samples) for the MRL samples. The T29 product had a 4.76% false-positive/negative rate (1 out of 21 samples) for the 0.25 MRL and 2 MRL samples. Finally, the T32 product demonstrated a 9.52% false-positive rate (2 out of 21 samples) for the 0.25 MRL samples and a 4.76% false-negative rate (1 out of 21 samples) for the MRL and 2 MRL samples.

## 4. Discussion

In the present study, we successfully established an efficient scientific verification and evaluation protocol. Colloidal gold immunochromatographic qualitative testing products from a total of 34 companies/manufacturers were evaluated and validated. If a product showed 0% false positives and 0% false negatives across all four groups of samples, it had 100% accuracy. Among the products tested for emamectin benzoate, several failed to meet the requirements. The product from T20 fell short of the false-negative/positive-rate requirement for both the blank and the MRL levels. Similarly, the product from T25 failed to meet the false-positive/negative-rate requirement at 2 MRL, whereas the product from T28 did not meet the requirement for the MRL level. For isocarbophos, the failures were more severe. The product from T30 missed the false identification rate requirement at the MRL level. The product from T31 showed deficiencies across multiple levels, failing to meet the false identification rate requirements for the blank, MRL, and 2 MRL levels. Additionally, the product from T33 underperformed at 0.25 MRL, MRL, and 2 MRL, demonstrating significant gaps in compliance. According to the results for fipronil, the product from T12 failed to meet the false identification rate requirement across all four tested groups, highlighting a widespread issue. The product from T22 fell short at the 0.25 MRL and MRL levels, whereas the products from T29 and T32 failed to meet the false identification rate requirement specifically at the 2 MRL level.

Overall, approximately 90% of the products passed the evaluation test and can be recommended for pesticide residue detection. Additionally, none of the companies/manufacturers had more than one failed product. The high overall accuracy and pass rate indicate that the manufacturing and development of rapid test products have reached relatively high levels.

Currently, pesticides are widely used in agriculture worldwide to protect crops from diseases and insects. However, since pesticides usually have direct contact with agricultural products, which travel from farm to fork, their negative effects on human health should not be ignored. Exposure to pesticides may cause foodborne illnesses, including cancer; reproductive diseases; failure of the immune or nervous system; and even death [14]. The establishment of MRLs is based on scientific risk assessment systems, and it is primarily used to regulate the use of legal but restricted pesticides. However, even when some pesticides are prohibited, MRLs are still needed to manage the potential risks they pose and ensure food safety. Owing to their chemical properties, which may persist in the environment, these prohibited pesticides may still re-enter the food chain through environmental pathways, leaving potential residues in food [15]. Therefore, MRLs need to be set to monitor and control these potential residues. In some other cases, prohibited pesticides may be detected in agricultural products because of illegal use. The setting of MRLs helps regulatory agencies detect and combat illegal use, ensuring the safety of agricultural products.

In recent years, experts and scientists have been working on researching and developing new technologies that enable people to discover and identify pesticide residues faster and more accurately to reduce the chance of being exposed to unsafe amounts of residues. Rapid qualitative detection of colloidal gold immunochromatographic products relies on the principle of competitive inhibition immunochromatography. During the chromatography process, the target analyte in the sample competes with colloidal-gold-labeled specific antibodies, inhibiting the binding of the antibody to the haptene–protein conjugate on the T line of the nitrocellulose membrane. This leads to changes in the color intensity of the T line. Via comparing the color intensity of the T line with the color intensity of the C line, a qualitative judgment of the target analyte in the sample can be made. This method is widely used in the rapid detection of pesticide residues.

As rapid detection technology is still being developed and popularized, the number of rapid test product manufacturers continues to increase. Data bridge market research revealed that the global market for food-testing kits is anticipated to experience a compound annual growth rate (CAGR) of 8.02% throughout the forecast period from 2022 to 2029. This future growth is attributed to heightened concerns over food safety, stemming from an increase in incidents of foodborne diseases and food fraud, in turn propelling the demand for food-testing kits during the projected timeframe [16]. Therefore, periodic evaluations of assay performance have become necessary for the better interpretation of test results, and the currently available products still need to be further improved to achieve lower deviation and higher overall accuracy. Research has revealed that a number of domestically manufactured rapid-testing kits for pesticide and veterinary drug residues found on the market exhibit discrepancies between their stated detection limits and actual performance as well as variable quality and inadequate interbatch consistency. Prior to employing these products as regulatory tools for quality and safety, it is essential to validate key parameters, including the limit of detection, false-positive rates, false-negative rates, and stability [17]. The development of a scientific and standardized evaluation system for assessing rapid detection methods and products, along with enhancing the regulation of commercial rapid detection tools, is crucial for improving the accuracy of test results and facilitating their application in production and regulatory contexts.

In fact, several evaluation systems and regulations have been established in other countries. For example, the United States and Canada use an official registration model for managing rapid detection products. The U.S. Food and Drug Administration (FDA) classifies in vitro diagnostic rapid detection kits under Section 513 of the Federal Food, Drug, and Cosmetic Act (FDCA), dividing them into three categories (classes I, II, and III) on the basis of risk level. Devices requiring premarket approval are classified as Class III, whereas those subject to general controls or moderate regulatory control fall under Class I or II. This section is essential for FDA oversight, ensuring that new devices, including those for food and drug safety testing, meet safety and efficacy standards before being sold on the market [18]. In Canada, Health Canada requires producers of rapid detection products used for medical, laboratory, industrial, educational, or research purposes to apply for a registration number under the Controlled Drugs and Substances Act and its regulations as well as the Food and Drugs Act. The official registration model involves the regulatory department reviewing a product’s technical documentation or the manufacturer’s qualifications without evaluating the quality or technical performance of the rapid detection products [19,20]. These regulations may provide great examples and references for the future development of rapid pesticide test product evaluation and validation systems in China, given China’s current system for the production, sale, and certification of rapid detection products. Hopefully, standardized methods for assessing the technical parameters of rapid detection products will become advanced enough to enhance consistent and reliable quality evaluation [21].

Through this evaluation experiment, a new concept has emerged for the testing of products with a percentage of false positives of only 0.25 MRL (T29 and T30 for emamectin benzoate; T06 and T22 for isocarbophos); these products have high accuracy and tend to have a higher sensitivity to lower doses of pesticide residues. Although they may not be ideal indicators for determining if a pesticide residue exceeds the MRL, they can be recommended to food companies or systems with higher food safety standards for pesticide residues. The critical detection level should be further evaluated and clarified.

## Figures and Tables

**Figure 1 foods-14-00478-f001:**
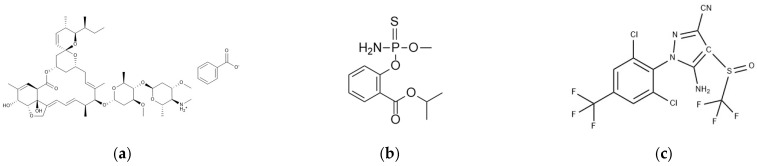
Structural formulas of emamectin benzoate (**a**), isocarbophos (**b**), and fipronil (**c**).

**Table 1 foods-14-00478-t001:** Detailed information on the pesticides.

Product Name	Formula	CAS#	Molecular Weight (g/mol)	MRL
Emamectin Benzoate	C_56_H_81_NO_15_	155569-91-8	1008.38	0.2 mg/kg
Isocarbophos	C_11_H_9_D_7_NO_4_PS	24353-61-5	289.29	0.05 mg/kg
Fipronil	C_12_H_4_Cl_2_F_6_N_4_OS	120068-37-3	437.15	0.02 mg/kg ^1^

^1^ The information in the table is from the GB 2763 National Standard [9].

**Table 2 foods-14-00478-t002:** Grouping and encoding rules for cowpea samples.

I (Blank)	II (0.25 MRL)	III (MRL)	IV (2 MRL)
Randomly generate 21 numbers from 1 to 84 for code assignment.	Randomly generate 21 numbers from 1 to 84, excluding the numbers in Group I, for code assignment.	Randomly generate 21 numbers from 1 to 84, excluding the numbers in Group I and II, for code assignment.	Randomly generate 21 numbers from 1 to 84, excluding the numbers in Group I, II, and III, for code assignment.

**Table 3 foods-14-00478-t003:** Relevant parameters of the evaluation index calculation formula.

Sample Status	Testing Result		Total
	Positive	Negative	
Positive	*TP*	*FN*	P_S_ = *TP + FN*
Negative	*FP*	*TN*	N_S_ = *TN + FP*
Total	P_R_ = *TP + FP*	F_R_ = *FN + TN*	T = P_R_ + F_R_ or P_S_ + N_S_
Sensitivity (%) = *TP*/P_S_ × 100 Specificity (%) = *TN*/N_S_ × 100 FN (%) = *FN*/P_S_ × 100 = 100 − Sensitivity FP (%) = *FP*/N_S_ × 100 = 100 − Specificity			

**Table 4 foods-14-00478-t004:** Requirements that need to be met for test products to obtain a result of ‘pass’.

	Blank	0.25 MRL	MRL	2 MRL
Specificity (%)	>90	>85		
FN (%)			≤5	0
FP (%)	≤10	≤15		

## Data Availability

The original contributions presented in the study are included in the article, and further inquiries can be directed to the corresponding author(s).

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
