# Peer review of "Evaluation and Validation of Colloidal Gold Immunochromatographic Qualitative Testing Products for the Detection of Emamectin Benzoate, Isocarbophos, and Fipronil in Cowpea Samples"

_foods, 2025, doi:10.3390/foods14030478_

Round 1

Reviewer 1 Report

Comments and Suggestions for Authors

The main problem with the paper is that the method applied are presented as black box. Instead, key characteristics of the test kists, such as principle of the measurement and how samples are evaluated and classified as positive or negative. One option would be to create an additional table giving all the relevant details of the test kits included in the study

·       Line 47, 48 ..Include reference of all guidelines applied in this study in the literature list. In addition, I don´t know, why the standard microbiology has been included in paper. I don´t see a link to the content of the paper

·       Table 4: Include all abbreviations in the captions

Author Response

1. The main problem with the paper is that the method applied are presented as black box. Instead, key characteristics of the test kits, such as principle of the measurement and how samples are evaluated and classified as positive or negative. One option would be to create an additional table giving all the relevant details of the test kits included in the study

Response: Thank you for reviewing our MS. We appreciate your insightful comment and fully agree with your suggestion. In response, we have added the principles of measurement and the classification rules for samples to the final paragraph of "2.2. Testing products" (lines 112–120). We kindly invite you to review the updated section.

2.Lines 47, 48 ..Include reference of all guidelines applied in this study in the literature list. In addition, I don´t know, why the standard microbiology has been included in paper. I don´t see a link to the content of the paper.

Response: Thank you very much for your feedback. References to all the guidelines applied in this study have been added to the citation list (Citations 2–4). Regarding the standard microbiology guideline mentioned in your review, I am uncertain if you are referring to the ISO guideline (Citation 1). The link to this guideline was not included as per the journal's citation format. However, the validation protocol is described within the guideline. If of interest, the guideline can be accessed at the following link:
ISO 16140-2:2016. Please let us know if further clarification is needed.

3.Include all abbreviations in the captions.

Response: Thank you. The abbreviation table has now been added (line 334).

Reviewer 2 Report

Comments and Suggestions for Authors

I suggest a major revision for this manuscript.

The chemical structures need to be redraw to get the same style for all.

The preparation of standards are not the part of sample preparation.

Clear description of sample preparation is missing.

Table 1 and 2 can be combined.

Why only these analytes were studied?

Any comparisons to HPLC/GC methods are missing.

How did the authors handle the false cases? An important paper may be cited here. Toth et al. Appl. Sci. 202212(23), 12005

Author Response

  1. The chemical structures need to be redrawn to obtain the same style for all.

Response: Thank you. The chemical structure of isocarbophos has been redrawn to match the style of the other two structures. Please kindly have a look.

  1. The preparation of standards is not part of sample preparation.

Response: Thank you. "The preparation of standard solution" has been revised and moved to the last paragraph of "2.1. Materials" (Lines 95--96).

  1. Clear description of sample preparation is missing..

Response: Thank you. We have added some details about the sample preparation to describe it more clearly (lines 122--126). The samples were weighed according to the operating requirements specified in each of the product instruction manuals after being verified via high-performance liquid chromatography‒tandem mass spectrometry.

  1. Tables 1 and 2 can be combined.

Response: Thank you very much. The tables are now combined.

  1. Why only these analytes were studied?

Response: Thank you very much for your feedback. We selected these analytes as they represent typical pesticide residues found on cowpea. Details about the analytes have been introduced and explained in the "Introduction" section. We kindly invite you to review this section.

  1. Any comparisons to HPLC/GC methods are missing.

Response: Thank you for your thoughtful comment. In response, we have added a brief comparison with HPLC methods in the "Introduction" section (lines 32–33). We chose not to elaborate further on HPLC methods, as it is commonly understood that they generally have lower overall efficiency. Instead, our focus remains on the principles and quality validation of rapid test products. We appreciate your understanding.

  1. How did the authors handle the false cases? An important paper may be cited here. Toth et al. Appl. Sci. 2022, 12(23), 12005.

Response: Thank you for the comment. For the handling of false cases, we mentioned that a certain rate of false positive/negative results in a "fail" for the evaluation, referring to the evaluation protocol from the European Union's guidelines and the Ministry of Agriculture. To be clearer, citations of the two guidelines were added to the citation list (citations 12,13).

Reviewer 3 Report

Comments and Suggestions for Authors

The work presents the possibility of using rapid screening tests to detect the presence of some pesticides in cowpea samples. These screenings would allow more controls at border control posts and rapid onsite tests with short times and low costs. However, these screenings need confirmation analysis using techniques such as mass spectrometry, in case of positivity. 

- what are the LODs? The various samples were spiked at three levels of addition: 0.25 * MRL, MRL and 2 * MRL, referring to MRLs for pesticide residues in food in China (GB 2763). Considering the EU Regulations that have lower MRLs, is this screening applicable? For example, Fipronil has a MRL 0.005 mg/kg with a legal definition that also takes into account its metabolite (Fipronil (sum fipronil + sulfone metabolite (MB46136) expressed as fipronil)).

- Add more explanation and a scheme with the mechanism of action of these Colloidal Gold Immunochromatographic Qualitative Testing Products.

- Can you add more information about the colour scale of the tests?  What about selectivity and specificity? Can you add pictures of the test strip cards as example?

- These screenings were tested for three target pesticides (Fipronil, Emamectin benzoate and Isocarbophos) for cowpea production. Were they tested for other pesticides or classes of pesticides?

- For each company, how many tests were performed?

- Did you test these screenings with other matrices or just cowpeas? Reals samples?

Author Response

  1. The work presents the possibility of using rapid screening tests to detect the presence of some pesticides in cowpea samples. These screenings would allow more controls at border control posts and rapid onsite tests with short times and low costs. However, these screenings need confirmation analysis using techniques such as mass spectrometry, in case of positivity. 

Thank you for reviewing our MS. We appreciate your insightful comment and fully agree with your suggestion. In response, we have added the screening process to clarify that that samples are free of the target pesticides(Lines 122-126). Please kindly review the updated section.

  1. what are the LODs? The various samples were spiked at three levels of addition: 0.25 * MRL, MRL and 2 * MRL, referring to MRLs for pesticide residues in food in China (GB 2763). Considering the EU Regulations that have lower MRLs, is this screening applicable? For example, Fipronil has a MRL 0.005 mg/kg with a legal definition that also takes into account its metabolite (Fipronil (sum fipronil + sulfone metabolite (MB46136) expressed as fipronil)).

Thank you for your insightful comment. In this study, we did not collect the exact LOD from each company as our evaluation scheme is setting up a protocol targeting the MRL stated in GB 2763 regulation. A product was determined as pass as long as it demonstrates the desired result under different MRLs at a high rate regardless of the LOD. Also, considering the products are developed for the pesticide residue detection in the foods in China as well as to make the study precise, we did not investigate the applicability of the products in other domain, such as the suitability for other regulations(like EU Regulations). With your suggestion, we will consider investigating the applicability for other regulations in later study as it would be a really good point to make the protocol and the products globalized.

  1. Add more explanation and a scheme with the mechanism of action of these Colloidal Gold Immunochromatographic Qualitative Testing Products.

Thank you for your comment. In response, we have added a brief explanation of the scheme with the mechanism (the principles of measurement and the classification rules for samples) to the final paragraph of "2.2. Testing products" (lines 112–120). We decided not to add much content on mechanism of action because the principles of colloidal gold immunochromatographic qualitative testing seems to be a common sense. We tend to focus more on the quality validation and evaluation protocol of those rapid test products.

  1. Can you add more information about the colour scale of the tests?  What about selectivity and specificity? Can you add pictures of the test strip cards as example?

Thank you. According to your suggestion, example pictures of the test strip card have been added (Figure A1-A3 in Appendix section). We have stated the calculation method of selectivity and specificity in table 3, and specificity is one of the important indicator to determine the evaluation result of the products (as stated in table 4).

  1. These screenings were tested for three target pesticides (Fipronil, Emamectin benzoate and Isocarbophos) for cowpea production. Were they tested for other pesticides or classes of pesticides?

Thank you. To make this study precise and easy to understand, we did not tested for other pesticides. As described in the “Introduction” section, we chose one matrix and three of the most typical pesticides that have a relatively high potential to be found on the matrix. We will continue to study on products targeting more matrix and pesticides in our later research.

  1. For each company, how many tests were performed?

Thank you for the question. As described in “2.3. Sample Preparation” section, 84 tests (21 tests for each group of MRL) were performed for each product from each company.

  1. Did you test these screenings with other matrices or just cowpeas? Reals samples?

Thank you for the question. The test was only performed on cowpea samples as the targets are typical pesticide residues having a potential to be discovered on cowpea crops in China. With the establishment of the protocol, we will expand and conduct more studies on evaluation of products targeting other food products.

Reviewer 4 Report

Comments and Suggestions for Authors

Dear Authors

The article “Evaluation and Validation of Colloidal Gold Immunochroma-tographic Qualitative Testing Products for the Detection of Emamectin Benzoate, Isocarbophos, and Fipronil in Cowpea Samples”, as the title mentioned, is about the evaluation and validation of 34 qualitative rapid testing products for the pesticide residues in cowpea (3 pesticides). Although the results of the manuscript are interesting, I have some suggestions and questions.

1.         Please, consider include the meaning of the abbreviations in all tables and in the Appendix (Tables 161 A1-A3).

2.         In the introduction the authors point out “An experiment was conducted in this study on cowpea samples for verification and evaluation of the quality of colloidal gold immunochromatographic qualitative rapid testing products for the pesticide residues mentioned above (emamectin benzoate, isocarbophos, and fipronil), with the aim of establishing scientific and standardized verification and evaluation protocols; therefore, the quality of commercially available rapid testing products can be ensured”…… therefore I consider that is important in the results highlight the established protocol for the scientific verification. On the other hand, it is only one protocol, then I suggest changing the meaning or the word "protocols" in the introduction.

3.         Also, I suggest that the authors highlight in the manuscript the scientific contribution of their study.

Author Response

  1. Please, consider include the meaning of the abbreviations in all tables and in the Appendix (Tables 161 A1-A3).

Response: Thank you. An abbreviation table has now been added (line 334).

  1. In the introduction the authors point out “An experiment was conducted ... therefore, the quality of commercially available rapid testing products can be ensured”…… therefore I consider that is important in the results highlight the established protocol for the scientific verification. On the other hand, it is only one protocol, then I suggest changing the meaning or the word "protocols" in the introduction.

Response: Thank you for your comment. According to your suggestion, we added the highlight "we successfully established an efficient scientific verification and evaluation protocol. " in "discussion" part (line 224-225). We also changed the word "protocols" to "protocol" in line 76 in the introduction.

  1. Additionally, I suggest that the authors highlight in the manuscript the scientific contribution of their study.

Response: Thank you for this comment. The scientific contribution was highlighted in the "Introduction" section(Lines 73-79). We kindly invite you to review the updated section.

Round 2

Reviewer 2 Report

Comments and Suggestions for Authors

The paper can be accepted.

Reviewer 3 Report

Comments and Suggestions for Authors

Thank you for your clarifications.

Supposedly, the LODs corresponde to the 0.25*MRL for the target analytes.